# Common pediatric surgical conditions and associated health-seeking behaviors in Pakistan: An urban and rural comparative assessment

Saqib Hamid Qazi[1], Syed Saqlain Ali Meerza[2], Sohail Lakhani[2], Sohail Asghar Dogar[1†], Zahra Ali Padhani[3], Mushtaq Mirani[2], Muhammad Khan Jamali[2], Sajid Muhammad[4], Muhammad Anwar[5], Saleem Islam[1], Sadaf Khan[6], Anjum Abbas[2], Zahid Ali Khan[2], Sana Khatoon[2], Imtiaz Sheikh[2], Rasool Bux[4], Rafey Ali[4], Hassan Naqvi[4], Arjumand Rizvi[4], Imran Ahmed Chahudary[4], Rizwan Haroon Ur Rashid[7], Syed Akbar Abbas[8], Abdul Sami Memon[9], Sadia Tabassum[10], Zara Bhatti[1], Abdur Rehman[2], Sajid Soofi[4], Jai K. Das[2,11]*

1 Section of Pediatric Surgery, Department of Surgery, Aga Khan University, Karachi, Pakistan, 2 Institute for Global Health and Development, Aga Khan University, Karachi, Pakistan, 3 University of Adelaide, Adelaide, Australia, 4 Center of Excellence in Women and Child Health, Aga Khan University, Karachi, Pakistan, 5 National Institute of Child Health, Karachi, Pakistan, 6 Section of General Surgery, Department of Surgery, Aga Khan University, Karachi, Pakistan, 7 Section of Orthopedic Surgery, Department of Surgery, Aga Khan University, Karachi, Pakistan, 8 Section of Otolaryngology & Head & Neck Surgery, Department of Surgery, Aga Khan University, Karachi, Pakistan, 9 Department of Ophthalmology, Aga Khan University, Karachi, Pakistan, 10 Section of Dermatology, Department of Medicine, Aga Khan University, Karachi, Pakistan, 11 Department of Paediatrics and Child Health, Aga Khan University, Karachi, Pakistan

† Deceased.
* jai.das@aku.edu

**Data Availability Statement:** All data of this research article is fully available in the paper and its Supporting Information files.

## Abstract

Approximately five billion people do not have access to necessary surgical treatment globally and up to 85% of children in LMICs are affected with conditions requiring surgical care by the age of 15 years. It is crucial to identify common surgical conditions in children in Pakistan to inform healthcare professionals and policymakers for effective resource allocation. This representative cross-sectional household survey conducted on children aged 5–10 years assessed existing surgical diseases and healthcare-seeking behaviors in the two largest provinces (Sindh and Punjab) of Pakistan. The data was collected through a validated cross-sectional survey tool [Surgeons OverSeas Assessment of Surgical Need (SOSAS)]. Caregivers were asked about their child's recent and past surgical conditions in six distinct anatomical regions and pictures were taken of identified conditions after appropriate consent for further diagnosis. Health-seeking behaviors including the kind of treatment sought, the nature of care received, and the reasons for not receiving care were noted. 13.5% of children surveyed reported a surgical condition, with a similar distribution across urban (13.2%) and rural (13.7) areas and the most common cause was trauma. The greatest number of surgical conditions were found to be on the head and neck region (57.7%), while the back accounted for the least number of conditions (1.7%). Our results outline a need for

**Funding:** This work was supported by the School Age Children Health and Nutrition Survey (SCANS) Consortium. (AGREEMENT/TVI & AKU/SCNAS/ 2020 to JKD) The funders had no role in study design, data collection and analysis, decision to publish, or preparation of the manuscript.

**Competing interests:** The authors have declared that no competing interests exist.

organizing all entities (governmental, non-governmental, and private) involved in child health to ensure efficient resource allocation to cater to existing surgical problems.

## Introduction

Conditions requiring surgical care are responsible for at least 11% of the overall global disease burden [1]. An estimated five billion people do not have access to basic surgical care, which makes the prevalence of surgical conditions in low- and middle-income countries (LMICs) particularly noteworthy [2]. In South Asia, more than 95% of the population lacks adequate access to medical care for conditions that require management through procedures [3]. Even more notable is the high surgical disease prevalence in children, with 85% of children in LMIC's having conditions requiring surgical care by the age of 15 years [4]. Many of these conditions manifest during the formative years of childhood, hence carry the possibility of lifelong disability or an increased risk of mortality [5] if not managed correctly in time.

In LMICs, where children make up nearly half of the overall population, there is relatively low attention paid to surgical conditions as the focus remains on malnutrition, infectious diseases and newborn care [6]. According to the United Nations, to attain the Sustainable Development Goals (SDGs), an approach towards health system strengthening and universal health coverage is essential to have access to appropriate healthcare [7]. The infrastructure, personnel, and resources necessary for pediatric surgical care are distinct from those required for adults [8]. As a crucial aspect of healthcare in LMICs, the enhancement of surgical healthcare provision for children is currently under focus [9] and this carries great socioeconomic benefit, as unresolved surgical issues potentiate medical expenses, incapacity, and loss of productivity. Therefore, techniques for improving pediatric surgical care delivery in low socio-economic areas can significantly lower childhood mortality and morbidity and decrease the associated psychosocial and financial stress [2, 10]. To achieve this goal, it is crucial to accurately identify the existing common pediatric surgical conditions, that can assist healthcare professionals and policymakers in efficient allocation of budget, skilled personnel, and equipment.

According to the available data from other low- and middle-income countries and national health indicators of Pakistan, the availability of general surgical care and anesthesia at facilities in Pakistan is estimated to be poor [11]. A small scale study in rural Pakistan has shown a high demand for pediatric surgical care, with a staggering 14.3% unmet operative need [12], thus highlighting the importance for a larger scale assessment. According to the World Bank, Pakistan, an LMIC in South Asia, is amongst the five most populous countries in the world [13] and children below the age of 18 years account for more than 45% of the population [14]. There are small scale studies that describe pediatric surgical crises in Pakistan, however, a large scale population representative study had yet to be conducted to gauge Pakistan's common pediatric surgical conditions and their distribution [15, 16] in a systematic manner. This study aims to evaluate the prevalence of pediatric surgical conditions in school age children and comprehend the healthcare-seeking patterns in rural and urban areas of the Punjab and Sindh provinces of Pakistan.

## Methodology

### Study design

This is a cross sectional household survey conducted on a population of children between the ages of 5–10 years, with the aim to outline the distribution of common surgical conditions and understanding the existing health seeking behavior.

## Study setting

The survey was carried out in the two largest provinces of Pakistan, Sindh and Punjab, which comprise more than two-thirds of Pakistan's population, in the year 2020–21. Punjab is the most densely populated province in Pakistan with a population of over 110 million [17] followed by Sindh with a population of approximately 47.9 million [18].

## Sample size and sampling strategy

The study was part of a larger survey focusing on the overall health and nutrition of school age children (5–10 years of age) [19, 20]. The sample size was estimated assuming the response rate of 90%, 95% confidence interval, 7% precision, 1.5% design effect, and the total minimum sample size required was 2055 individuals in Sindh and 3582 in Punjab. We took the Pakistan Bureau of Statistics framework and randomly selected enumeration blocks (clusters) form both the urban and rural areas (population proportionate), which were termed Primary Sampling Units (PSUs). Multistage sampling was done to select enumeration blocks and 20 households were randomly selected (Secondary Sampling Units) from each cluster. Households that refused to participate were replaced with the next house in the randomization process.

## Survey tool

Data was collected through a validated cross-sectional survey tool [Surgeons OverSeas Assessment of Surgical Need (SOSAS)]. SOSAS has evolved into a validated population-based household survey which evaluates the prevalence of conditions requiring surgical treatment [21] [22]. The SOSAS questionnaire is divided into two specific components; the first component focuses on demographic information and includes the age and gender of all the residents in a household (household members were defined as people living in the same residence) and the second component focuses on children's recent and past surgical conditions in six distinct anatomical regions: head, eyes, face, ears and neck; chest; back; abdomen; groin, genitals, and buttocks; arms, hands, legs, and feet. The nature of surgical condition whether acquired (example: wounds, hernia, renal stones, and bleeding per penis/rectum) or congenital (example: congenital cardiac and eye problems) and health seeking behaviors, including the kind of treatment sought, the nature of care received, and the reasons for not accessing or receiving care were all noted.

## Data collection

We conducted the survey in two rounds. Round one in Sindh was from (5th to 22nd March 2020) and round two was from (1st June - 17th July 2020). In Punjab round one was between (27th February-22nd March 2020) and round two from (19th September to 22nd October 2020). We collected data from all the six divisions and 27 districts in Sindh, and all the nine divisions and 36 districts in Punjab. Households were line-listed and a total of 5,677 (rural: 3,531 and urban: 2,146) eligible houses were randomized (**Table 1**). Research personnel who conducted the survey were trained for data collection and were monitored by senior research personnel. For initial assessment and improvements, a pilot study was conducted in 50 households. Information was recorded electronically through an application developed specially for this survey by the Data Management Unit (DMU). An application was developed for Android, and an IIS-10 (the latest version of Internet Information Services (IIS), which is an extensible web server developed by Microsoft [23]) webserver was set up with MySQL (MySQL Database is a server system that includes multithreaded SQL servers that support different back ends, client programs and libraries, administrative tools, and a range of application-programming

**Table 1. Details of children with lesions.**

| | | Rural | Urban |
|---|---|---|---|
| Clusters in Sindh | | 51 | 57 |
| Clusters in Punjab | | 123 | 67 |
| Total households | | 3531 | 2146 |
| Total number of children | | 5,073 (63.2%) | 2,953 (36.8%) |
| Total number of children with surgical lesions | | 585 (11.5%) | 350 (11.9%) |
| Total number of surgical lesions | | 1,020 (66.4%) | 516 (33.6%) |
| Socioeconomic status of family | High income | 222 (38.1%) | 265 (75.9%) |
| | Low income | 362 (62%) | 84 (24.1%) |
| School attendance | Yes | 454 (77.7%) | 302 (86.5%) |
| | No | 130 (22.3%) | 47 (13.5%) |
| Age | 5 years | 166 (16.3%) | 82 (15.9%) |
| | 6 years | 238 (23.4%) | 115 (22.3%) |
| | 7 years | 212 (20.8%) | 124 (24.0%) |
| | 8 years | 218 (21.4%) | 106 (20.5%) |
| | 9 years | 185 (18.2%) | 89 (17.2%) |
| Gender of the child | Male | 2,616 (51.6%) | 1,503 (50.9%) |
| | Female | 2,457 (48.4%) | 1,450 (49.1%) |

interfaces [24]) database to collect data in remote areas. The application was used on Samsung Tab A7 tablet. To restrict inappropriate access to the application, authorized data collectors required to authenticate using login credentials which were specifically generated for the use of this application on the server. Along with the survey, a set of 736 pictures of surgical conditions were taken, from a total of 1,536 (47.9%), after obtaining written informed consent from patients and caregivers. These pictures were then shared with relevant doctors at the Aga Khan University Hospital (AKUH) for further diagnosis, and reconfirmation. Pictures were taken through the same application and stored on the server under password protection.

## Data analysis

Surgical conditions in children, either present at the time of the survey or in the past were noted as conditions requiring surgical care. Unmet surgical need was defined as a surgical condition that requires active surgical care, for which the child did not access or receive surgical care. Sampling weights were added to the data at household and individual level, to account for unequal selection probabilities and non-response. A standard survey module was used to consider the multi-stage survey design including stratification, clustering, and sampling weights. Descriptive statistics for the subjects were estimated and reported as mean (±SD), median, ranges and frequencies as appropriate. The analyses estimated results at district level with population subgroups such as age, gender, school status (in-school or out-of-school) and geographical location (urban or rural), and districts of the region. The study households were divided into five socio-economic strata and the lowest two were categorized as 'low income' and the upper three as 'high-income'. Descriptive analysis was performed using STATA 16 (Stata Statistical Software: Release 16. College Station, TX: StataCorp LP) [25]. Chi squared tests were performed to determine the association between a child's school attendance and socioeconomic status of the family, and the nature of surgical condition. A p-value of less than 0.05 was considered statistically significant to conclude an association.

### Ethics statement

Ethical approval was granted by the Ethics Review Committee (ERC) of the Aga Khan University and the National Bioethics Committee (NBC). At every stage of the data handling process, protecting the privacy of all gathered information (data and pictures) was ensured and every interview was conducted in a confidential setting and encrypted data was secured in password protected central database. The research participants were informed about the purpose, methods and benefits and intended use of the research. Informed consent was obtained from the parent/guardian of each participant under 18 years of age. Respondents were free to stop the interview at any time or skip any questions they did not want to answer. They had the right to ask questions at any point before, during, or after the interview. All interviews were conducted by trained staff and in conditions of privacy. In cases where necessary, we informed all the households of a surgical diagnosis requiring attention as identified through the pictures.

## Results

### Demographics

Of the 5,677 households, we surveyed a total of 8,026 children, of which 935 children were noted to have surgical conditions that required attention. (**Table 1**).

### Surgical conditions identified

**Lesions.**   A total of 585 (11.5%) children in rural, and 350 (11.9%) children in urban areas were identified to have a total of 1,536 surgical conditions and few children had more than one surgical condition (**Table 1**). Pictures were taken of 736, after appropriate consent, for further diagnosis and review (**S1 Table**).

Through a head-to-toe survey via the SOSAS tool, the head and neck region (comprising of the head, eye, ear, face, and neck) wase the most frequently affected region in both rural (54.3%) and urban (65.9%) areas and the common conditions were wounds secondary to trauma (**Table 2**).

**Head and neck.**   A total of 886 surgical conditions were identified in this region and the most affected was the face (27.7%), followed by the head (25.8%), ears (20.9%), eyes (20.9%), and neck (4.7%). Conditions noted on the head, eyes and face were mostly secondary to trauma, while conditions on ears and neck were consistent with acquired conditions such as mass or growth or congenital deformities.

**Table 2. Lesions on head and neck.**

| Region | Eyes (n = 185) | | Ears (n = 185) | | Face (n = 245) | | Head (n = 229) | | Neck (n = 42) | |
|---|---|---|---|---|---|---|---|---|---|---|
| Surgical problem | Lesions n (%) | | Lesions n (%) | | Lesions n (%) | | Lesions n (%) | | Lesions n (%) | |
|  | Urban | Rural | Urban | Rural | Urban | Rural | Urban | Rural | Urban | Rural |
| Wound (Secondary to accident) | 10 (12.6) | 30 (28) | 4 (7.0) | 8 (4.8) | 75 (75.8) | 117 (76.5) | 65 (64.4) | 89 (60.0) | 1 (9.6) | 2 (4.4) |
| Wound (Others) | 13 (20.8) | 18 (12.6) | 41 (64.6) | 74 (58.5) | 7 (11.4) | 20 (15.3) | 9 (15.9) | 29 (23.3) | 2 (9.4) | 9 (30.0) |
| Burn | 1 (0.8) | 4 (4.1) | 1 (0.8) | 0 0 | 2 (2.3) | 2 (1.2) | 2 (1.9) | 1 (0.7) | 2 (13.9) | 1 (5.6) |
| Mass or growth (Solid) | 5 (5.6) | 5 (4.2) | 2 (2.7) | 3 (2.2) | 1 (0.6) | 1 (0.9) | 4 (3.3) | 9 (9.6) | 4 (33.3) | 12 (48.0) |
| Deformity congenital | 16 (20.4) | 29 (24) | 13 (21.2) | 19 (14.9) | 5 (5.7) | 5 (2.5) | 1 (0.7) | 2 (1.3) | 4 (28.6) | 3 (10.2) |
| Deformity acquired | 2 (2.7) | 3 (3.0) | 1 (2.5) | 7 (9.5) | 3 (4.2) | 7 (3.7) | 11 (13.8) | 7 (5.1) | 1 (5.4) | 1 (1.8) |
| Blindness | 0 (0) | 1 (0.8) | N/A | N/A | N/A | N/A | N/A | N/A | N/A | N/A |
| Reduced Vision | 23 (37.1) | 25 (23.5) | N/A | N/A | N/A | N/A | N/A | N/A | N/A | N/A |
| Hearing loss | N/A | N/A | 1 (1.3) | 11 (10.1) | N/A | N/A | N/A | N/A | N/A | N/A |

(The percentages in the table below are a result of a weighted analysis)

A total of 328 pictures were taken of the surgical conditions and identified scars/keloids (n = 76) and scarring alopecia (n = 47) as the most common conditions on the head, and post-traumatic disfiguring scars (n = 42) on the face. Growths, comprising of hemangiomas (n = 4) and mass/cyst/swellings (n = 4) were the most common conditions on the neck. Care was not sought for 6.7% conditions in the rural, and 11.7% in urban settings. (**Table 2**).

**Chest.** A total of 48 conditions were identified and majority of the conditions in rural areas were secondary to traumatic injury (20.4%), closely followed by non-injury-related wounds (19.8%) including congenital deformities (**Table 3**). Falls were the major reasons for traumatic surgical conditions in both urban (81.3%) and rural (75.8%) areas.

A total of 11 pictures identified pectus excavatum (n = 5) and lymphangioma (n = 3) as the most commonly presenting conditions.

**Back.** We identified 26 conditions on the back and majority consisted of non- traumatic lesions, including congenital deformities and acquired mass or growths (**Table 3**).

The 12 pictures identified hemangiomas (n = 4) and other soft tissue swelling (n = 2). In urban settings, burn injury (25.3%) was the most common condition. care was not sought for 14.5% conditions in rural and 38.6% in urban areas.

**Abdomen.** Of the 106 surgical conditions noted in the abdomen, most consisted of abdominal distention with pain and children's inability to urinate appropriately (**Table 3**). A set of 40 lesion pictures identified abdominal hernia (n = 22) as the most common surgical condition of the abdomen and most wounds secondary to trauma resulted from burns, in both urban (50.8%) and rural (48.4%) areas.

**Buttocks/ groin/ genitalia.** Among a total of 49 surgical conditions, most consisted of non- traumatic lesions in both urban and rural areas and congenital deformities and mass/ swellings were the common conditions (**Table 3**).

A total of 19 pictures identified inguinal hernia (n = 7) and gluteal abscesses (n = 6) as common conditions. Of the wounds secondary to trauma, falls were noted as the most common cause in both urban (48.8%) and rural (72.5%) areas.

**Table 3. Lesions on chest, back, abdomen, buttocks/groin/genitalia.**

| Region | Chest (n = 48) | | Back (n = 26) | | Abdomen (n = 106) | | Buttocks/Groin/Genitalia (n = 49) | |
|---|---|---|---|---|---|---|---|---|
| Surgical problem | Lesions (%) | | Lesions (%) | | Lesions (%) | | Lesions (%) | |
| | Urban | Rural | Urban | Rural | Urban | Rural | Urban | Rural |
| Wound (Secondary to accident) | 2 (14.4) | 7 (20.4) | 0 (0) | 4 (19.5) | 1 (3.0) | 3 (3.1) | 1 (5.3) | 1 (2.7) |
| Wound (Other) | 3 (12.1) | 6 (19.8) | 2 (38.6) | 7 (39) | 3 (8.6) | 4 (7.2) | 4 (20.1) | 12 (29.2) |
| Burn | 1 (3.3) | 1 (3.0) | 2 (25.3) | 1 (2.7) | 2 (11.3) | 5 (7.1) | 0 (0) | 2 (9) |
| Mass or growth (Solid) | 3 (18.1) | 0 (0) | 1 (15.7) | 3 (11.8) | 1 (1.3) | 2 (6.5) | 3 (22.7) | 2 (3.4) |
| Mass or growth (Reducible) | 0 (0) | 0 (0) | 0 (0) | 0 (0) | 0 (0) | 2 (1.8) | 1 (15) | 2 (10.4) |
| Deformity congenital | 4 (29.5) | 4 (14) | 1 (20.4) | 4 (22.4) | 0 (0) | 4 (5.6) | 2 (9.1) | 6 (16.3) |
| Deformity acquired | 0 0 | 3 (10.8) | 0 (0) | 1 (4.6) | 0 (0) | 0 (0) | 1 (2.4) | 3 (12.1) |
| Foreign body | 2 (6.2) | 5 (17.7) | 0 (0) | 0 (0) | 0 (0) | 1 (1.4) | 0 (0) | 0 (0) |
| Cardiac (Congenital) | 3 (16.4) | 4 (14.1) | 0 (0) | 0 (0) | 1 (2.6) | 2 (3.2) | N/A | N/A |
| Abdominal distention or pain | N/A | N/A | N/A | N/A | 10 (33.2) | 16 (21.1) | N/A | N/A |
| Inability to urinate | N/A | N/A | N/A | N/A | 10 (25.7) | 27 (34) | 0 (0) | 0 (0) |
| Renal stone | N/A | N/A | N/A | N/A | 2 (11.2) | 2 (1.6) | 0 (0) | 0 (0) |
| Bleeding (Per Rectum) | N/A | N/A | N/A | N/A | 1 (3.1) | 7 (7.5) | 2 (10.3) | 3 (11.8) |
| Bleeding (Per Penis) | N/A | N/A | N/A | N/A | N/A | N/A | 1 (15) | 2 (5.3) |

(The percentages in the table below are a result of a weighted analysis)

**Table 4. Lesions on extremities.**

| Region | Fingers (n = 32) | | Thumb/Hand (n = 48) | | Upper arm (n = 49) | | Lower arm (n = 73) | | Upper leg (n = 24) | | Lower leg (n = 73) | | Foot (n = 122) | |
|---|---|---|---|---|---|---|---|---|---|---|---|---|---|---|
| **Surgical problem** | Lesions (%) | | Lesions (%) | | Lesions (%) | | Lesions (%) | | Lesions (%) | | Lesions (%) | | Lesions (%) | |
| | Urban | Rural | Urban | Rural | Urban | Rural | Urban | Rural | Urban | Rural | Urban | Rural | Urban | Rural |
| Wound (Secondary to accident) | 5 (70.5) | 17 (69.9) | 6 (29.5) | 11 (30.5) | 4 (25.7) | 18 (45.7) | 10 (73.7) | 34 (54.2) | 2 (42.4) | 6 (26) | 11 (48.5) | 21 (42.3) | 17 (48.1) | 40 (41.8) |
| Wound (Other) | 0 (0) | 2 (10.7) | 1 (6.1) | 8 (25.7) | 0 (0) | 12 (39.5) | 0 (0) | 11 (22.6) | 1 (21.1) | 3 (13.3) | 2 (10.2) | 12 (26.6) | 3 (8) | 19 (23.4) |
| Burn | 0 (0) | 1 (1.9) | 2 (19.4) | 6 (19.7) | 1 (13.9) | 2 (3.9) | 1 (5) | 2 (2.9) | 2 (36.5) | 2 (14.4) | 8 (21.2) | 3 (6.6) | 5 (15.3) | 12 (15.2) |
| Mass or growth (Solid) | 0 (0) | 0 (0) | 2 (8.4) | 0 (0) | 0 (0) | 1 (3) | 1 (4.3) | 1 (3.6) | 0 (0) | 1 (3) | 0 (0) | 0 (0) | 0 (0) | 0 (0) |
| Deformity congenital | 0 (0) | 2 (9.4) | 2 (22.5) | 1 (5.8) | 0 (0) | 0 (0) | 1 (7.1) | 1 (1.7) | 0 (0) | 2 (12.1) | 0 (0) | 3 (4.5) | 2 (5.4) | 10 (12.1) |
| Deformity acquired | 3 (29.5) | 2 (8.1) | 1 (14.2) | 6 (13.8) | 8 (60.4) | 3 (7.8) | 2 (9.9) | 9 (15) | 0 (0) | 4 (23.8) | 4 (20.1) | 6 (10.1) | 4 (18.7) | 8 (7.7) |
| Recurrent drainage / discharge | 0 (0) | 0 (0) | 0 (0) | 2 (4.6) | 0 (0) | 0 (0) | 0 (0) | 0 (0) | 0 (0) | 1 (7.4) | 0 (0) | 3 (10.1) | 2 (4.4) | 0 (0) |

(The percentages in the table below are a result of a weighted analysis)

**Extremities.** Out of the total 202 conditions; 11.9% were on the upper arm, 36.1% on lower arm, and 39.6% on thumb/hand, and fingers. Majority of the conditions were secondary to trauma (falls) in in both rural and urban areas. Of the 156 pictures taken, post-burn severe depigmentation (n = 12) and lacerations/big scars (n = 12) were identified as common surgical conditions.

Out of the 219 surgical conditions on the lower limbs, 10.6% were on upper leg, 33.3% on lower leg, and 55.7% on feet and lesions were mostly due to trauma. A total of 170 pictures identified lacerations and scars (n = 16) as the most common conditions. A high percentage (26.5% in rural and 28.1% in urban) of population did not access surgical care (**Table 4**).

## Healthcare seeking behavior

Approximately 86.8% of surgical conditions in urban, and 86.3% in rural areas were managed at a health care facility, ranging from private, government, and NGO-based health facilities. The remaining 13.7% surgical conditions in rural and 13.2% in urban areas received no surgical care, or were managed by a traditional healer. A small fraction of surgical conditions identified received only medical management despite visiting a health care facility (4.7% in rural areas and 3.5% in urban areas).

The main reasons identified for not seeking surgical care included the caregiver's perception of the condition being non-surgical [rural areas (48.5%), and urban areas (55.4%)], followed by an inability to afford health care. Approximately 33.6% patients and their families in rural areas could not afford health services, which was 8.9% higher than the patient population in urban areas. (**Table 5**).

Comparison of the type of surgical injuries and regions of the body affected was done between children who attended school and those who did not attend school. We found that children who did not attend school were more likely to have surgical wounds as a result of falls (P-value 0.02) and there was no statistically significant difference in the region of the body affected between school going and non-school going children, more details in **Table 6**.

A comparison was done between children belonging to low and high income socioeconomic groups for the type of wounds which required surgical care, and the region of the body affected. Children belonging to low socioeconomic groups were found to be affected more by traumatic

**Table 5. Lesion management and impact.**

| | | | Rural | Urban | p-value | Low Income | High Income | p-value |
|---|---|---|---|---|---|---|---|---|
| Total number of children with surgical conditions | | | 585 | 350 | | 435 | 500 | |
| Total number of surgical conditions | | | 602 (100%) | 312 (100%) | | 451 (100%) | 463 (100%) | |
| lesion managed at Health care facility | Yes | | 492 (82.0%) | 267 (83.5%) | 0.6596 | 367 (81.1%) | 392 (83.9%) | 0.3668 |
| | No | | 110 (18.0%) | 45 (16.5%) | | 84 (18.9%) | 71 (16.1%) | |
| | Reasons for not visiting health facility | No money for health care | 39 (28.9%) | 15 (30.1%) | 0.9159 | 40 (39.2%) | 14 (18.2%) | 0.0389 |
| | | No money for transportation | 3 (2.2%) | 0 (0.0%) | | 3 (3.0%) | 0 (0.0%) | |
| | | No time (Died before arrangements) | 4 (3.8%) | 1 (4.1%) | | 2 (2.8%) | 3 (5.1%) | |
| | | Fear / No trust | 5 (4.1%) | 3 (3.3%) | | 2 (2.0%) | 6 (6.0%) | |
| | | Not available (Facility/personnel/ equipment) | 7 (4.4%) | 3 (4.6%) | | 4 (3.0%) | 6 (6.1%) | |
| | | No need (Condition is not surgical) | 61 (54.5%) | 25 (52.0%) | | 36 (45.8%) | 50 (62.6%) | |
| | | Other | 3 (2.3%) | 2 (5.5%) | | 3 (4.3%) | 2 (2.1%) | |
| Where were these conditions managed? | At home | | 33 (6.2%) | 7 (2.5%) | 0.3184 | 16 (4.1%) | 24 (5.8%) | 0.1643 |
| | Government health facility | | 214 (45.6%) | 119 (43.9%) | | 170 (47.4%) | 163 (42.7%) | |
| | Private health facility | | 190 (38.3%) | 118 (45.7%) | | 135 (37.4%) | 173 (43.9%) | |
| | NGO health facility | | 3 (1.4%) | 2 (0.9%) | | 4 (2.0%) | 1 (0.4%) | |
| | Hakeem/Traditional Healer | | 52 (8.6%) | 21 (6.9%) | | 42 (9.0%) | 31 (7.2%) | |
| Type of treatment received for surgical conditions | None/No surgical care | | 12 (2.7%) | 4 (1.3%) | 0.4595 | 6 (1.9%) | 10 (2.6%) | 0.7897 |
| | Only medical treatment | | 251 (53.2%) | 125 (50.1%) | | 187 (53.5%) | 189 (51.3%) | |
| | Major procedure (A procedure which requires regional or general anesthesia) | | 22 (4.8%) | 18 (6.7%) | | 16 (4.3%) | 24 (6.5%) | |
| | Minor procedure (Dressings, wound care, punctures, suturing and I&D) | | 153 (28.9%) | 93 (31.1%) | | 114 (29.0%) | 132 (30.2%) | |
| | Manipulation / casting / sling | | 43 (8.4%) | 25 (9.8%) | | 36 (9.7%) | 32 (8.0%) | |
| | Traction | | 11 (2.0%) | 2 (0.7%) | | 8 (1.7%) | 5 (1.5%) | |
| Does this problem (Disability) still impact on child's daily life (Multiple response) | The condition is not disabling | | 530 (87.4%) | 276 (88.7%) | 0.6536 | 395 (87.7%) | 411 (88.0%) | 0.9346 |
| | He/she feels ashamed | | 39 (6.6%) | 13 (3.3%) | | 31 (7.0%) | 21 (4.0%) | |
| | He/she not able to work like he/she used to | | 21 (3.5%) | 11 (3.5%) | | 14 (2.9%) | 18 (4.1%) | |
| | He/she needs help with transportation | | 24 (4.0%) | 7 (2.3%) | | 20 (4.4%) | 11 (2.6%) | |
| | He/she needs help with daily living | | 21 (4.2%) | 15 (5.4%) | | 16 (4.1%) | 20 (5.0%) | |

wounds secondary to stab/slash/cut/crush (P-value 0.00), bite or animal attacks (P-value 0.02) and falls (P-value 0.02). The face (P-value 0.04) was the most affected region (**Table 6**).

## Discussion

This large-scale survey identifies common surgical conditions and the care received in children between the ages of 5–10 years, in the two most densely populated provinces of Pakistan. This

**Table 6. Subgroup analysis according to school attendance and socioeconomic status.**

| Surgical problem type | Children attending school (%) | | | | Socioeconomic status (%) | | | |
|---|---|---|---|---|---|---|---|---|
| | Yes (n = 4632) | No (n = 945) | Total | P-value | Low (n = 3358) | High (n = 4668) | Total | P-value |
| **Injury/accident-related wound** | | | | | | | | |
| I. Car, truck, bus crash | 0.036 | 0 | 0.03 | 0.55 | 0.047 | 0.015 | 0.028 | 0.32 |
| II. Motorcycle crash | 1.06 | 0.62 | 0.99 | 0.27 | 0.99 | 0.79 | 0.88 | 0.37 |
| III. Pedestrian, bicycle crash | 0.15 | 0 | 0.12 | 0.30 | 0.14 | 0.1 | 0.12 | 0.65 |
| IV. Gunshot/firearm | 0.033 | 0 | 0.028 | 0.66 | 0.046 | 0.014 | 0.027 | 0.36 |
| V. Stab/slash/cut/crush | 0.88 | 1.16 | 0.92 | 0.45 | 1.2 | 0.48 | 0.78 | 0.00* |
| VI. Bite or animal attack | 0.26 | 0.052 | 0.22 | 0.09 | 0.44 | 0.17 | 0.28 | 0.02* |
| VII. Fall | 3.17 | 5.06 | 3.48 | 0.02* | 4.08 | 2.9 | 3.39 | 0.02* |
| VIII. Open fire/explosion | 0.14 | 0.062 | 0.13 | 0.45 | 0.082 | 0.15 | 0.12 | 0.41 |
| IX. Hot liquid / hot object | 0.24 | 0.17 | 0.23 | 0.66 | 0.25 | 0.18 | 0.21 | 0.51 |
| **Region of surgical problem** | | | | | | | | |
| Head | 2.56 | 2.85 | 2.6 | 0.63 | 2.96 | 2.44 | 2.66 | 0.27 |
| Eyes | 2.87 | 2.36 | 2.78 | 0.48 | 2.47 | 2.72 | 2.61 | 0.53 |
| Ears | 2.1 | 2.72 | 2.2 | 0.26 | 2.61 | 1.81 | 2.15 | 0.06 |
| Face | 2.96 | 3.47 | 3.05 | 0.43 | 3.46 | 2.41 | 2.84 | 0.04* |
| Neck | 0.66 | 0.53 | 0.64 | 0.65 | 0.54 | 0.53 | 0.54 | 0.93 |
| Chest | 0.67 | 0.71 | 0.68 | 0.91 | 0.65 | 0.72 | 0.69 | 0.74 |
| Back | 0.38 | 0.51 | 0.4 | 0.57 | 0.34 | 0.33 | 0.34 | 0.94 |
| Abdomen | 1.35 | 1.85 | 1.43 | 0.25 | 1.4 | 1.43 | 1.42 | 0.91 |
| Buttocks/Groin/Genitalia | 0.58 | 0.97 | 0.65 | 0.18 | 0.68 | 0.59 | 0.63 | 0.65 |
| Extremities | 0.022 | 0 | 0.019 | 0.64 | 0.016 | 0.02 | 0.013 | 0.34 |

was a population representative survey to understand the surgical disease burden in children and helped assess the nature of surgical conditions, severity, site distribution, along with variance by urban and rural divide and socio-economic status in Pakistan.

The survey highlights a high surgical burden, with a similar distribution across both urban and rural areas of Pakistan. The highest number of surgical conditions were found on the head and neck region (57.7%) while the back region accounted for the least number of surgical conditions (1.7%). The survey identified a high percentage of surgical conditions for which children did not receive any surgical care. Calculated as an average of urban (13.2%) and rural (13.7%), this amounted to a staggering 13.5%. Across both rural and urban areas, the extremities were identified to have the highest percentage of surgical conditions for which care was not received (26.5% and 28.1% respectively) due to multiple reasons. Of the caregivers who did not seek healthcare services for their child's lesion, approximately half believed that the condition did not require surgical attention. This is high in comparison to a recent smaller scale study in a rural district of Pakistan, where the unmet operative need was calculated as 14.3% [12]. This study conducted at a larger scale highlights a similar pattern of high pediatric surgical disease burden in Pakistan. In comparison to other South Asian countries including Nepal (unmet surgical need of 5%) [26] and our neighboring country India (unmet surgical need of 6.5%) [27] the surgical lesions having not received appropriate care in Pakistan is more than double (13.5%), pointing towards the urgent need for attention by policy makers. The number of physicians in the province of Punjab are 0.83 per 1000 people [28], in Sindh there are 0.53 per 1000 people [29] and in overall Pakistan, there are 1.1 physicians per 1000 people and [30] it can be safely assumed that the number of surgeons would be even low and hence signifying a higher current unmet surgical need.

Different behavioral models exist to understand human behavior; we attempt to understand the healthcare usage in the study through the Anderson Healthcare Utilization Model. It attempts to understand the determinants of health services utilization, to assess inequality in access to health services, and to facilitate the policy-making process for equitable access to care and health services [31]. This model highlights three factors in health care utilization which includes predisposing factors, enabling factors and need. Our hypothesized predisposing factors that led to decreased healthcare usage in the survey included the study population being children, with decreased health and treatment related knowledge, decreased healthcare related literacy in caregivers. Lack of enabling factors such as healthcare insurance (decreased financial capacity) and decreased general availability of healthcare information. The need being variable, but for children that did not receive care, most caregivers did not perceive that the existing condition required medical attention. This brings to notice the need for improvement measures in capacity building, community engagement, resource allocation and policy development to improve surgical care access in especially the rural areas of Pakistan.

This study provides evidence that majority of the surgical conditions noted affect the head, face and neck region of children, which not only requires timely surgical treatment but also needs to be followed up for post-trauma psychosocial effects [32]. Community programs, similar to existing programs for communicable diseases like diarrhea and pneumonia, are required [33]. However, work needs to be done in expansion and incorporation of surgical disease awareness, to increase the local population's knowledge of common operative conditions and promote capacity building of health facilities at the district level. It is vital to spread a better understanding of the various types of surgical conditions and when to seek timely care, in order to reduce the associated morbidity and mortality in children. To prevent possibly disabling problems in children, awareness and training campaigns on preventative strategies for common causes of injuries like burns, falls and electrocution should be undertaken. Policies for health care subsidies are required in rural areas where most families are unable to afford health care for their children, especially as this study highlighted the inability to afford health care services as the second most common reason for not seeking care. Additionally, a large percentage of surgical conditions were secondary to trauma, so investing in lay-person training [34] for pre-hospital basic and preventive care can help reduce the burden on local health care facilities and can also provide the critical extra time needed for transportation of patients to hospitals.

Further efforts are required to identify health care delivery resources at local health facilities with respect to the patient population visiting those centers. Regular assessment of resources and trained personnel, and intercommunication systems between local community hospitals may allow for efficient and timely redistribution of resources as per need and in an efficient manner. There is also a need to assess existing training for surgical trauma care at facilities and accordingly, further training and implementation of trauma care guidelines [35] can help streamline the process of surgical health care delivery, in effect assisting emergency staff in making quicker decisions leading to effective acute surgical care.

Potential limitations of this study include the existing inter-provincial differences in healthcare delivery setup (Punjab having the most developed health care delivery system and Baluchistan and Khyber Pakhtunkhwa having the least developed [36]). However, it does offer significant representation of any rural and urban setting in Sindh and Punjab. Another limitation is the survey's in-built definition of surgical conditions, as the count of operative conditions in this survey relied upon caregiver's report of the condition in a verbal interview. Hence, the numbers identified via this survey serve as proxies in estimating the burden of surgical conditions. The survey also does not directly take into account chronic diseases such as visceral malignancies, which could require surgical care, thus underestimating the true surgical

burden of disease. Despite these limitations, operative conditions recognized by the SOSAS survey provides an estimate for the urgent need to improve surgical consultations and care in Pakistan.

## Conclusion

The common surgical conditions and the associated high percentage of lesions not receiving appropriate surgical care requires attention of government policy makers. Our results have outlined the urgent need for organizing all entities (governmental, non-governmental and private) involved in pediatric surgical care delivery in both urban and rural areas. Policies at the federal, provincial and district government level are required for the efficient allocation of resources, skilled personal (doctors and nurses), capacity building and awareness programs on surgical diseases and urgent care which should be planned, implemented, and assessed at the community level.

## Supporting information

**S1 Table. Diagnoses of photographs taken at the study site.** Table A in S1 Table head, Table B in S1 Table eyes, Table C in S1 Table ears, Table D in S1 Table face, Table E in S1 Table neck, Table F in S1 Table chest, Table G in S1 Table back, Table H in S1 Table abdomen, Table I in S1 Table buttocks/groin/genitalia, Table J in S1 Table extremities.
(DOCX)

## Acknowledgments

We would like to acknowledge the support of the community in both the provinces who participated and made this study possible. We would thank the support of the administrative team at Aga Khan University and Trust for Vaccines and Immunization who took care of all the logistics needed to conduct a study of this scale.

We would like to take this opportunity to dedicate this study to our dear colleague and friend Dr Sohail A Dogar who was a key member of this study team. He unceremoniously left us for the heavenly abode on 31[st] July 2024. Though he is no longer with us, his dedication and compassion would continue to inspire and guide our research.

## Author Contributions

**Conceptualization:** Saqib Hamid Qazi, Jai K. Das.

**Data curation:** Syed Saqlain Ali Meerza, Muhammad Khan Jamali, Muhammad Anwar, Anjum Abbas, Zahid Ali Khan, Sana Khatoon, Imtiaz Sheikh, Rasool Bux, Rafey Ali, Hassan Naqvi, Arjumand Rizvi, Imran Ahmed Chahudary, Syed Akbar Abbas, Sadia Tabassum, Zara Bhatti, Abdur Rehman.

**Formal analysis:** Syed Saqlain Ali Meerza, Sohail Lakhani, Abdur Rehman.

**Funding acquisition:** Saqib Hamid Qazi, Zahra Ali Padhani, Jai K. Das.

**Methodology:** Syed Saqlain Ali Meerza.

**Project administration:** Zahra Ali Padhani, Mushtaq Mirani, Sajid Soofi, Jai K. Das.

**Resources:** Saqib Hamid Qazi, Syed Saqlain Ali Meerza, Sohail Asghar Dogar, Saleem Islam, Rizwan Haroon Ur Rashid, Abdul Sami Memon, Sajid Soofi, Jai K. Das.

**Software:** Sajid Muhammad.

**Supervision:** Saqib Hamid Qazi, Syed Saqlain Ali Meerza, Mushtaq Mirani, Saleem Islam.

**Validation:** Syed Saqlain Ali Meerza.

**Visualization:** Saqib Hamid Qazi.

**Writing – original draft:** Syed Saqlain Ali Meerza.

**Writing – review & editing:** Saqib Hamid Qazi, Syed Saqlain Ali Meerza, Sadaf Khan.

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
