## [Decision Letter · Decision Letter 0]

2 Jul 2024

PGPH-D-24-01152

Common pediatric surgical conditions and associated health-seeking behaviors in Pakistan: an urban and rural comparative assessment

Dear Dr. Das,

Thank you for submitting your manuscript to PLOS Global Public Health. After careful consideration, we feel that it has merit but does not fully meet PLOS Global Public Health’s publication criteria as it currently stands. Therefore, we invite you to submit a revised version of the manuscript that addresses the points raised during the review process.

EDITOR: Please insert comments here and delete this placeholder text when finished. Be sure to:

Indicate which changes you require for acceptance versus which changes you recommendAddress any conflicts between the reviews so that it's clear which advice the authors should followProvide specific feedback from your evaluation of the manuscript

Please ensure that your decision is justified on PLOS Global Public Health’s publication criteria and not, for example, on novelty or perceived impact.

We look forward to receiving your revised manuscript.

Kind regards,

Dhananjaya Sharma, MS, PhD, DSc, FRCS

Academic Editor

Journal Requirements:

2. Please amend your Data Availability Statement and indicate where the data may be found.

Additional Editor Comments (if provided):

The manuscript has been acknowledged as important by both reviewers but they have suggested minor revision

Please see their comments and revise accordingly and resubmit with point-vise response

Reviewers' comments:

Reviewer's Responses to Questions

**Comments to the Author**

1. Does this manuscript meet PLOS Global Public Health’s publication criteria? Is the manuscript technically sound, and do the data support the conclusions? The manuscript must describe methodologically and ethically rigorous research with conclusions that are appropriately drawn based on the data presented.

Reviewer #1: Yes

Reviewer #2: Yes

2. Has the statistical analysis been performed appropriately and rigorously?

Reviewer #1: I don't know

Reviewer #2: Yes

3. Have the authors made all data underlying the findings in their manuscript fully available (please refer to the Data Availability Statement at the start of the manuscript PDF file)?

Reviewer #1: Yes

Reviewer #2: Yes

4. Is the manuscript presented in an intelligible fashion and written in standard English?

Reviewer #1: Yes

Reviewer #2: Yes

5. Review Comments to the Author

Reviewer #1: This study asks a good question, what are the most common surgical issues in children in Pakistan, comparing rural and urban in Sindh and Punjab, and what are associated behaviors. This is definitely a research gap, given the population dynamics in Pakistan and healthcare system resource allocations.

The language is good, manuscript is written well and is easy to understand. The title is representative of the study, and abstract and conclusions are informative. Good references are provided.

Comments for authors to consider:

This survey may show differences in Sindh and Punjab, especially differences in rural Sindh and rural Punjab. Table 1 indicates the urban vs rural clusters in Sindh and Punjab, but the data is combined after that. Separating information from these two groups may highlight significant differences and add to the information provided in this paper.

The authors find almost half the cases that did not seek care felt it was not needed. Does the survey support this based on the interview and photos reviewed by experts?

Pakistan has more doctors/1000, can the authors comment on Punjab vs Sindh, and urban vs rural differences in the clusters surveyed?

The authors discuss the importance of their findings in detail but I find some statements generalized rather than being focused on specific areas based on the survey findings.

- The discussion regarding care at health centers (tertiary, district via referral systems) (lines 333 to 343) seems out of place based on this survey. As the authors themselves pointed out, Pakistan has more doctors, and there is no significant difference between overall unmet surgical needs in children in urban and rural setting. The authors have not provided any information regarding density of doctors in the surveyed clusters, or an objective assessment of access to hospitals, clinics, and emergency rooms.

- Based on Table 3, most of the children were treated at a healthcare facility (> 80 % in each group), but most received only medical treatment (> 50 % in each group). The authors mention trauma care guidelines. A better discussion about this finding will be helpful.

Punjab has a more developed healthcare system compared to other provinces in Pakistan. The study may underrepresent the overall picture, as Baluchistan and KPK findings may be quite different. Did the authors consider this as a limitation of their study?

Authors noted percentages in the tables, does their analysis show any statistical differences that might be of interest to the reader, especially in Table 3?

Appreciate the analysis presented in Table 4 which includes P values.

Reviewer #2: Dear Authors,

Thank you for a well-written manuscript. The topic is a very important one and I commend you on taking this up. Please see my comments/suggestions as a peer reviewer.

1) Line 58 - instead of density could you use "prevalence"?

2) Line 65 - 'which include "health system....universal health coverage"'. This is not an explicitly mentioned as an SDG. The reference you quote is an interpretation of what is needed to achieve some of the SDGs. I recommend you change the sentence to reflect this.

3) Line 71 - 'improving the standard of pediatric surgical care'. I recommend that you change it to "improving...surgical care delivery" since that is the gist of your study. Surgical care usually stands for technical surgical skills.

4) Lines 76-80 - This is good material. Could you add any literature on availability of non pediatric surgical care? it would be good to position your work in that context.

5) Line 84 - I suggest changing 'health-seeking' to 'healthcare seeking'.

6) Under Methodology, and elsewhere, please change to active voice.

7) Line 93 - change 'comprised' to 'comprise'.

8) Lines 97,98 - A reference to the original (larger) survey is needed here. I would also encourage a 2-3 sentence description of that larger survey.

9) Line 109 - The use of SOSAS concerns me. Why was this chosen? From perusing the site it appears a group involved primarily in providing surgical mission trips- a model that has been criticized for not aiming for native capacity development. Your study is looking at creating long-term strategies for surgical care delivery. Perhaps another tool would have been more useful? Work by Debas et.al (Tanzania), Stewart et al (Ghana) come to mind.

- I also recognize that you cannot change a survey tool after the study has been done, and the work you have done is very important; I recommend adding information where SOSAS was used and why you chose it. It would be good to add the limitations and advantages of using SOSAS in the discussion section.

10) Line 121-126 - It occurs to me that this survey was conducted during the peak of the COVID-19 first waves (with its concomitant social distancing regulations?) . Mentioning this in the discussion section, along with the implications for sampling, and where surgical issues (emergent and non- emergent) where in the respondents radar would be good.

11) Line 130, 131 - Please indicate "To make the survey available on mobile platforms...", it is likely that your readers, as was the case with this reviewer, will not be familiar with the acronyms that you mention here. It would help immensely if you attach references or explain the acronyms.

12) Line 134- 138 - This part really concerned me and left me confused. From the description of the study it appeared that this was a study of perceived need for surgical care; the collection of photos seems to be an attempt at diagnosis- is this appropriate for the study? Was the Ethics Review Board aware of this? If a diagnosis was mode were the patients referred to appropriate care. I understand the challenge of doing a field level survey of surgical conditions and would like a better description of why a 'diagnostic' approach was taken.

13) Line 165 - did you survey children or their parent/guardian?

14) Line 172, 173 - What happens after the diagnosis is made? ( see 12 above). Pediatric populations are considered vulnerable populations in most situations. Was the diagnosis communicated back to the patient/family with directionso f next steps?

15) Line 170- 247- this section is very elaborate and well written. However, I found myself losing the plot as I read this. With the many tables here, I would encourage you to decrease the written text.

16) Line 248 - I suggest changing this to "Healthcare Seeking Behavior".

17) Line 300, 301 - It would be good to reflect on the original survey to convey that this was a representative sample, prior to indicating the rural and urban rates.

18) In the discussion section I would recommend, if possible, a framework of looking at healthcare seeking behavior to interpret the results of this study. There are elegant models (Anderson, Gelberg etc) that have looked at this. The Gelberg model would perhaps be a good one to interpret the results with.

Again, I commend you on this important, and I am sure a very labor intensive, study. Thank you for undertaking it.

6. PLOS authors have the option to publish the peer review history of their article (what does this mean?). If published, this will include your full peer review and any attached files.

**Do you want your identity to be public for this peer review?** For information about this choice, including consent withdrawal, please see our Privacy Policy.

Reviewer #1: **Yes: **Uzma Khan

Reviewer #2: **Yes: **Shailendra Prasad

---

## [Decision Letter · Decision Letter 1]

6 Aug 2024

PGPH-D-24-01152R1

Common pediatric surgical conditions and associated health-seeking behaviors in Pakistan: an urban and rural comparative assessment

Dear Dr. Das,

Thank you for submitting your manuscript to PLOS Global Public Health. After careful consideration, we feel that it has merit but does not fully meet PLOS Global Public Health’s publication criteria as it currently stands. Therefore, we invite you to submit a revised version of the manuscript that addresses the points raised during the review process.

We look forward to receiving your revised manuscript.

Kind regards,

Dhananjaya Sharma, MS, PhD, DSc, FRCS

Academic Editor

Journal Requirements:

Additional Editor Comments (if provided):

Minor revision as adv by reviewer 1

Reviewers' comments:

Reviewer's Responses to Questions

**Comments to the Author**

1. If the authors have adequately addressed your comments raised in a previous round of review and you feel that this manuscript is now acceptable for publication, you may indicate that here to bypass the “Comments to the Author” section, enter your conflict of interest statement in the “Confidential to Editor” section, and submit your "Accept" recommendation.

Reviewer #1: All comments have been addressed

Reviewer #2: All comments have been addressed

2. Does this manuscript meet PLOS Global Public Health’s publication criteria? Is the manuscript technically sound, and do the data support the conclusions? The manuscript must describe methodologically and ethically rigorous research with conclusions that are appropriately drawn based on the data presented.

Reviewer #1: Yes

Reviewer #2: Yes

3. Has the statistical analysis been performed appropriately and rigorously?

Reviewer #1: I don't know

Reviewer #2: Yes

4. Have the authors made all data underlying the findings in their manuscript fully available (please refer to the Data Availability Statement at the start of the manuscript PDF file)?

Reviewer #1: Yes

Reviewer #2: Yes

5. Is the manuscript presented in an intelligible fashion and written in standard English?

Reviewer #1: Yes

Reviewer #2: Yes

6. Review Comments to the Author

Reviewer #1: Thank you for revising the manuscript.

I have a few minor comments.

Line 33: Recommend change sentence to “This representative cross-sectional household survey conducted on children aged 5-10 years assessed existing surgical diseases and healthcare-seeking behaviors in the two largest provinces (Sindh and Punjab) of Pakistan.”

Line 41: Recommend change sentence to “13.5% of children surveyed reported a surgical condition, with a similar distribution across urban (13.2%) and rural (13.7) areas.”

Line 44- Please add a sentence about trauma being the most common underlying reason for the conditions.

Ethics statement: Please add a sentence regarding handling of photos, and communication of diagnosis and treatment recommendations to study participants. This would answer any questions raised in the minds of the readers, similar to concerns raised by a reviewer.

Table 3- Recommend using “Low Income” and “High Income”, rather than rich and poor. The authors may want to define the income thresholds used for these categories in the methodology section.

When discussing findings reported in Table 4, use (Table 4) rather than saying “more details in Table 4”.

Reviewer #2: Thank you for addressing my questions. I appreciate the enormous work that has gone into this!

7. PLOS authors have the option to publish the peer review history of their article (what does this mean?). If published, this will include your full peer review and any attached files.

**Do you want your identity to be public for this peer review?** For information about this choice, including consent withdrawal, please see our Privacy Policy.

Reviewer #1: No

Reviewer #2: **Yes: **Shailendra Prasad

---

## [Decision Letter · Decision Letter 2]

23 Aug 2024

Common pediatric surgical conditions and associated health-seeking behaviors in Pakistan: an urban and rural comparative assessment

PGPH-D-24-01152R2

Dear Dr. Das,

We are pleased to inform you that your manuscript 'Common pediatric surgical conditions and associated health-seeking behaviors in Pakistan: an urban and rural comparative assessment' has been provisionally accepted for publication in PLOS Global Public Health.

Best regards,

Dhananjaya Sharma, MS, PhD, DSc, FRCS

Academic Editor

All comments have been satisfactorily addressed

Reviewer Comments (if any, and for reference):

Reviewer's Responses to Questions

**Comments to the Author**

1. If the authors have adequately addressed your comments raised in a previous round of review and you feel that this manuscript is now acceptable for publication, you may indicate that here to bypass the “Comments to the Author” section, enter your conflict of interest statement in the “Confidential to Editor” section, and submit your "Accept" recommendation.

Reviewer #1: All comments have been addressed

2. Does this manuscript meet PLOS Global Public Health’s publication criteria? Is the manuscript technically sound, and do the data support the conclusions? The manuscript must describe methodologically and ethically rigorous research with conclusions that are appropriately drawn based on the data presented.

Reviewer #1: Yes

3. Has the statistical analysis been performed appropriately and rigorously?

Reviewer #1: Yes

4. Have the authors made all data underlying the findings in their manuscript fully available (please refer to the Data Availability Statement at the start of the manuscript PDF file)?

Reviewer #1: Yes

5. Is the manuscript presented in an intelligible fashion and written in standard English?

Reviewer #1: Yes

6. Review Comments to the Author

Reviewer #1: Appreciate the authors reviewing and modifying the manuscript

7. PLOS authors have the option to publish the peer review history of their article (what does this mean?). If published, this will include your full peer review and any attached files.

**Do you want your identity to be public for this peer review?** For information about this choice, including consent withdrawal, please see our Privacy Policy.

Reviewer #1: No
